# BECLIN1: Protein Structure, Function and Regulation

**DOI:** 10.3390/cells10061522

**Published:** 2021-06-17

**Authors:** Sharon Tran, W. Douglas Fairlie, Erinna F. Lee

**Affiliations:** 1Olivia Newton-John Cancer Research Institute, Heidelberg, VIC 3084, Australia; Sharon.Tran@onjcri.org.au; 2School of Cancer Medicine, La Trobe University, Bundoora, VIC 3086, Australia; 3La Trobe Institute for Molecular Science, La Trobe University, Melbourne, VIC 3086, Australia

**Keywords:** BECLIN1, PI3K Class III complexes, autophagy, BCL-2

## Abstract

BECLIN1 is a well-established regulator of autophagy, a process essential for mammalian survival. It functions in conjunction with other proteins to form Class III Phosphoinositide 3-Kinase (PI3K) complexes to generate phosphorylated phosphatidylinositol (PtdIns), lipids essential for not only autophagy but other membrane trafficking processes. Over the years, studies have elucidated the structural, biophysical, and biochemical properties of BECLIN1, which have shed light on how this protein functions to allosterically regulate these critical processes of autophagy and membrane trafficking. Here, we review these findings and how BECLIN1’s diverse protein interactome regulates it, as well as its impact on organismal physiology.

## 1. Introduction

Autophagy is defined as the process by which intracellular entities undergo lysosomal degradation. Its execution relies on a large network of molecules that co-operate to maintain cellular homeostasis due to its protein and organelle turnover function. Autophagy induced under environmental challenges such as hypoxia or starvation is also an adaptive response that helps cells to survive these conditions as a result of the bioenergetic nature of the degradation reaction, and through the use of breakdown products in other metabolic pathways. Although many proteins regulate autophagy, in this review, we will specifically focus on BECLIN1, an evolutionarily conserved protein that is a well-recognised positive regulator of this process due to its critical role in the biogenesis of autophagosomes, the double-membrane vesicles that enclose degradation targets and fuse with lysosomes.

### BECLIN1 Discovery and Function

The first ortholog of BECLIN1 discovered was yeast *ATG6/VPS30*, identified in two separate screens for gene products that could regulate autophagy and the yeast vacuolar protein sorting pathway [1,2]. The mammalian counterpart, BECLIN1, was later discovered in a screen for proteins interacting with the anti-apoptotic protein B cell lymphoma 2 (BCL-2) [3]). Genetic analysis revealed BECLIN1 to be a candidate tumour suppressor, monoallelically deleted in 40–75% of sporadic breast and ovarian cancers [4]. Subsequently, several gain- and loss-of-function studies across a range of cell types and organisms confirmed BECLIN1 to be a haploinsufficient tumour suppressor [5,6,7], but also demonstrated BECLIN1 to be important across diverse physiological settings, including in murine embryo development [5,6], dauer development, immunity, and neuronal and cardiac health [8]. In addition to these autophagy-attributed physiological roles, BECLIN1 has also been shown to regulate endocytic trafficking and LC3-associated phagocytosis (LAP) and is essential in Drosophila for wing and blood development and salivary secretion [9,10], worm intestinal function and embryo development [11], and murine neuronal health [12], skin development [13], and intestinal barrier function [14].

BECLIN1 performs both of its autophagy and membrane trafficking functions by interacting with several other proteins—primarily vacuolar protein sorting-associated protein 15 (VPS15), VPS34, UV radiation resistance-associated gene product (UVRAG), and autophagy-related protein 14 (ATG14). Together, these assemble into two different Class III PI3K complexes, Complex 1 (C1) and Complex 2 (C2), depending on whether ATG14 or UVRAG respectively is present. Within these complexes, the catalytic lipid kinase subunit, VSP34, is responsible for the phosphorylation of PtdIns, which then mediates autophagy and/or membrane trafficking functions through the recruitment of effector proteins [15].

Based on structural, biochemical, and functional studies, it has become apparent that BECLIN1 serves as a scaffolding protein for these two complexes, functioning as a signal integration hub for different cellular pathways to allosterically regulate the lipid kinase activity of VPS34. In the sections below, we aim to review the structure and function of BECLIN1, its regulation by its protein interactome, and how these interactions impart downstream effects, particularly on lipid kinase activity, and any known associated physiological effects of this regulation.

## 2. BECLIN1 Protein Structure and Binding Interactions

Human BECLIN1 is a 450-amino-acid protein with three major functional domains: (1) an extended N-terminal region (residues 1–150) that is intrinsically disordered and contains multiple phosphorylation sites [16,17] plus a well-characterised BCL-2 homology-3 (BH3) domain required for interactions with BCL-2 proteins [18]; (2) a coiled-coil (CC) domain (residues 174–266) that interacts with the CC domains of UVRAG or ATG14; and (3) the β-α autophagy-specific (BARA) domain (residues 266–450) that engages with membranes. Part of the overlapping region between the C-terminal end of the CC domain and the BARA domain is referred to as the Evolutionarily Conserved Domain (ECD), though it is sometimes referred to interchangeably with the BARA domain due to the high degree of overlap between these regions. In addition, a region between the BH3 and the CC domains (residues 141–171) was found to adopt helical conformations and is termed the Flexible Helix Domain (FHD, [19]). Although essential for starvation-induced autophagy [19], to date, limited characterisation of this FHD region exists.

In the absence of UVRAG or ATG14, there is evidence that BECLIN1 homodimerises through homotypic interactions involving its CC domain [20,21,22]. A crystal structure of the homodimerised BECLIN1 CC domain shows that it consists of two antiparallel helices packed together with “imperfect” residue pairings that cause it to be metastable in this form [21]. However, mutually exclusive binding to either UVRAG or ATG14 via CC domain interactions results in the formation of more stable dimers. Interestingly, the binding affinity between BECLIN1-UVRAG dimers is greater than for BECLIN1-ATG14 dimers, possibly favouring the formation of the BECLIN1-UVRAG complex (C2, discussed further below) and its associated activities, at the expense of the BECLIN1-ATG14 complex (C1) and its activities [21]. Several structures of BECLIN1 in the context of the PI3K C1 and C2 complexes have been determined and will also be discussed below.

The ECD/BARA domain consists of three β-sheets and α-helices located at the C-terminal region of BECLIN1 [23]. There appear to be two main functions of this region: it is the site of BECLIN1 membrane interaction and engages with the VPS34 lipid kinase. Mutation of the ECD region results in a loss of VPS34 immunoprecipitation, autophagy deficiency, and a lack of tumour suppressor function in xenograft models [24]. The lipid binding capabilities are specifically attributed to the presence of three amino acids that make up an aromatic finger motif—F359, F360, and W361 [25]. More recently, it was found that the negative regulator of autophagy, Run domain protein as BECLIN1 interacting and cysteine-rich containing (RUBICON, discussed further below), exerts its function through interaction with the BECLIN1 BARA domain [26]. Here, it was speculated that RUBICON binding could prevent BARA β-sheet unfolding and engagement with membranes, resulting in reduced VPS34 activity [26].

## 3. BECLIN1 Is a Subunit of PI3K Class III Complexes

As mentioned above, BECLIN1 is a scaffold in the PI3K Class III C1 and C2 complexes and allosterically regulates its associated lipid kinase activity. Both complexes are composed of the three core proteins VPS34, VPS15, and BECLIN1, but differ in their fourth core subunit and fifth subunit/accessory proteins [27,28,29,30].

In C1, the unique fourth core subunit is ATG14, with nuclear receptor binding factor 2 (NRBF2) sometimes included as a fifth accessory protein. The C1 complex is primarily involved in autophagy, specifically in the nucleation of the autophagosomal double membrane (Figure 1a, [27,28,29,30]). In experiments using ATG14 as a surrogate marker for the complex, C1 localises to autophagosomes and their biogenesis sites at the endoplasmic reticulum (ER, [27,28,29,30,31]), providing localised phosphorylation of PtdIns before dissociating from the nascent phagophore membrane [32]. It should, however, be noted that ATG14 also has C1-independent functions outside of these early nucleation stages of autophagy. Without a requirement for the other C1 subunits including BECLIN1, ATG14 interacts with the t-SNARE complex (STX17 and SNAP29) and SNARE effector protein Snapin to facilitate autophagosome–lysosomal fusion and endocytic trafficking activity, respectively [33,34,35]. Thus, care is needed when interpreting results using PI3K subunits as a representative of the entire complex, given the evidence for individual subunits of C1 functioning without requirement for other binding partners. In general, the additional subunit NRBF2 appears to enhance ATG14-linked VPS34 activity (as discussed further below, [36,37]); however, conflicting reports on this activity do exist [38].

In C2, the unique fourth core subunit is UVRAG, with two additional subcomplexes formed due to the binding of either RUBICON or BAX interacting factor 1 (BIF-1/Endophilin B1) as the fifth accessory protein (Figure 1a). The role of C2 is more enigmatic than C1. In addition to autophagy [39], C2 is reported to play a role in endocytic trafficking [12,27] and lysosomal tubulation [40]. Increased VPS34 activity and pro-autophagic outcomes are also associated with the BIF-1 subcomplex of C2 [41], as well as autophagy-independent activities including EGFR signal regulation and cytokinesis [42]. In contrast, the presence of RUBICON in C2 has a negative effect on both autophagy and endocytic trafficking [28,30]. In addition to the aforementioned disruption of membrane binding by the BECLIN1 BARA domain [26], this effect of RUBICON is suggested to be due to its role as a negative regulator of Rab7 activation, which is important for both autophagosome and endosome maturation [43], plus a role in binding and inhibiting VPS34 [44]. UVRAG, similarly to ATG14 in C1, also appears to have C2-independent functions without a requirement for BECLIN1 interaction. It can interact with the membrane fusion complex c-VPS, to stimulate the maturation of both autophagosomes and endosomes (i.e., their fusion with lysosomes, [45]). Taken together, this suggests that BECLIN1 is essential to C1 and C2 for the nucleation of autophagosomes and endosomes, but it is not fully understood if it is crucial for all members of the PI3K complexes to be present in order to direct the later steps of autophagosome and endosomal maturation.

Structural determination of human and yeast C1 and C2 reveals similar overall three-dimensional conformations between the two complexes. Both complexes are dynamic and take on a “V” shape, with VPS34 and VPS15 located on one catalytic arm of the “V” and BECLIN1 and either ATG14 or UVRAG on the other regulatory arm of C1 and C2, respectively [46,47,48,49]. In contrast to earlier studies [24], these structural studies proposed that there is no direct contact between VPS34 and BECLIN1 but, instead, that this is mediated by VPS15 [46,48]. Another difference between these studies and earlier studies that suggested anti-parallel coiled-coil alignments between BECLIN1 and either ATG14 or UVRAG [21] is that the coiled-coil organisation is actually parallel in the C1/C2 complexes [46,48]. The BECLIN1 BARA domain containing the aromatic finger motif sits at the start of the “V” arm, interacting with the ATG14 C-terminal domain (where the Barkor/ATG14L autophagosome targeting sequence (BATS) domain is) or the UVRAG BARA2 domain [47,48]. Although not clearly resolved, the base of the “V” consists of both the N-terminus of BECLIN1, which receives the majority of the autophagy-stimulating phosphorylation events, and the N-termini of ATG14 or UVRAG. Three disconnected helices were able to be modelled in the yeast C2 N-terminal base region, demonstrating an interaction with VPS15 [48]. It is unclear if these comprise the BH3 helix and FHD helix. However, together, these structures suggest that the tips of the “V” mediate membrane binding, whereas the apex is a regulatory hub, explaining how several phosphorylation events taking place in the BECLIN1 BARA domain could sterically hinder membrane binding (Section 4.3). The overall structural similarity between C1 and C2 also suggests that it is unlikely for conformational reasons to drive functional differences between the two complexes. Instead, this could be due to different membrane binding properties. Accordingly, the ATG14 BATS domain at the regulatory arm tip is reported to be responsible for C1 membrane binding to PtdIns-containing liposomes and purified ER in a much stronger manner than UVRAG in C2 [47].

In addition to RUBICON, structural information has been determined for the BECLIN1 complexes containing NRBF2 and how this accessory protein influences C1 activity. These studies suggest that human NRBF2 binds the base of C1, interacting with a helix that is tentatively assigned to BECLIN1 [50,51,52]. Depending on their stoichiometric ratios, homodimers of NRBF2 may enable a symmetric heterotetramerisation of C1 and NRBF2 [50] and binding of NRBF2 to C1 can liberate the activation loop of VPS34 kinase domain so that it becomes poised for catalytic action [52].

## 4. The BECLIN1 Protein Interactome and How It Regulates Downstream Processes

In addition to its important role in the two Class III PI3K complexes, BECLIN1 also interacts with numerous other proteins from various cellular pathways and is even targeted by viruses and viral homologues [53]. These interactions can impart multiple post-translational modifications (PTMs), including phosphorylation, ubiquitination, ISGylation, acetylation, and cleavages. As BECLIN1 serves as a scaffold within the PI3K complexes, these PTMs have the ability to allosterically mediate secondary effects on PI3K complex activities. In this section, we examine the relationship between BECLIN1 and these other interacting proteins, their associated PTMs, and how these interactions govern autophagic activity.

### 4.1. Regulation of Autophagy via BECLIN1:BCL-2 Family Interactions

As mentioned above, BECLIN1 was first identified as a BCL-2 interacting partner [3]. In addition to the historic significance of this discovery, the interaction between BECLIN1 and the anti-apoptotic BCL-2 family of proteins is an archetype for how diverse signalling pathways and proteins can converge on either protein to modulate PI3K activity, autophagy, and apoptosis. BCL-2 has a “sequestering” action on BECLIN1, where binding between the two reduces available BECLIN1 levels for the formation of C1 and C2 and thereby inhibits autophagy [54]. Many kinases and proteins have been discovered to target BECLIN1 or BCL-2, influencing their binding affinities and thus promoting or reducing autophagy in ways that affect diverse physiological processes including the maintenance of cardiac health, neurodegeneration, and longevity.

BECLIN1 binds to members of the BCL-2 family via its BH3 domain. The crystal structures of peptides corresponding to this region of BECLIN1 in complex with pro-survival proteins BCL-2 and BCL-XL mirror the binding mode of pro-apoptotic BH3-only proteins (Figure 2a). Here, the BECLIN1 BH3 domain forms an amphipathic helix that binds to the conserved hydrophobic groove of the pro-survival proteins [55,56,57]. This interaction is the primary site of crosstalk between these two critical cellular processes of autophagy and apoptosis; however, it is unclear if BECLIN1 influences apoptosis through binding to BCL-2. At least one study has shown that, unlike other BH3-only proteins, BECLIN1 does not have the same neutralising effects on BCL-2 [58], probably due to the lower affinity of the BECLIN1 BH3 domain compared to that of the pro-apoptotic BH3-only proteins [56,57]. In contrast, members of the BCL-2 family—BCL-2, BCL-XL, MCL-1, and BCL-B [54,59,60]—have all been shown to influence autophagy by binding to and inhibiting BECLIN1 activity to downregulate autophagy.

Not surprisingly, interplay between BECLIN1, BCL-2, and other proteins can dictate autophagy levels. Competitive binding of proteins to BECLIN1 or BCL-2 can disrupt or reinforce the BECLIN1:BCL-2 interaction (Figure 2b), with promotion of autophagy as a result of displacing BECLIN1 from BCL-2. For instance, the ER transmembrane protein VMP-1 displaces BCL-2 to facilitate its own interaction with BECLIN1 and localise BECLIN1 to the ER, where autophagosomes are generated [62]. The BECLIN1:BCL-XL complex is also displaced by enforced expression of the ARF tumour suppressor via an unknown mechanism [63]. In contrast, factors that instead enhance the BECLIN1:BCL-2 interaction inhibit BECLIN1-mediated autophagy in various cell lines. These factors include nutrient-deprivation factor 1 (NAF-1, [64]), inositol 1,4,5-trisphosphate receptor (IP_3_R, [65]), and reticulon 3 (RTN3, [66]). Another competitively binding protein that promotes autophagy is activating molecule in BECLIN1 regulated autophagy (AMBRA). Under basal conditions, AMBRA tethers BECLIN1 to the cytoskeleton through an interaction with the dynein motor complex [67]. Autophagy-stimulating conditions that result in ULK1-dependent phosphorylation of AMBRA release it and BECLIN1 from the cytoskeleton, allowing it to relocate to autophagosome biogenesis sites [67]. AMBRA can be sequestered by BCL-2 but, under autophagy-inducing conditions, is released to instead bind BECLIN1 [68,69]. Similarly, the pro-apoptotic BH3-only protein BIM, which also interacts with BCL-2, mediates a second BECLIN1 cytoskeletal link. Here, BIM suppresses autophagy by mislocalising BECLIN1 to the dynein motor complex.

In addition to competitive displacement by binding partners, the BECLIN1:BCL-2 protein interactions can also be disrupted or reinforced by phosphorylation events in the BECLIN1 BH3 domain to sterically disrupt these complexes (Figure 3). Phosphorylation of BECLIN1 at T119 by the tumour suppressors death-associated protein kinase 1 and 2 (DAPK1/2, [70,71]) or rho kinase 1 (ROCK1, [72]) disrupts the BECLIN1:BCL-2 interaction, leading to increased autophagy. In contrast, phosphorylation of BECLIN1 at T108 by MST1 mediates the opposite effect and enhances the interactions, though the basis of how this occurs has been questioned following structural analysis of the phosphorylated BECLIN1 BH3- BCL-2 protein complexes [56,73]. Phosphorylation of BECLIN1 might also have secondary implications in altering its affinity for additional BCL-2-related protein modulators. For example, Bag-1 (BCL-2 associated athanogene-1), a heat-shock protein co-chaperone with multiple cellular functions, interacts with the T119-phosphorylated form of BECLIN1 [74]. Alternatively, phosphorylation of BCL-2 by c-JUN N-terminal kinase 1 (JNK1) reduces its interaction with BECLIN1 [75]. An interesting case is high mobility group box 1 (HMGB1), which both directly binds to BECLIN1 and promotes BCL-2 phosphorylation via extracellular signal regulated kinase (ERK), promoting dissociation to induce autophagy via two distinct mechanisms [76].

Regulation of autophagy by modulation of this BECLIN1:BCL-2 interaction is implicated in general organismal health span and is potentially a therapeutically targetable avenue for the treatment of cancer or cardiac dysfunction. Mice engineered with knock-in mutations of BECLIN1 (F121A) that have reduced BCL-2 binding display increased autophagy in tissue (including skeletal muscle, kidney, and heart). Attributed to these increases in autophagy activities are the phenotypes of decreased amyloid accumulation in the brain [77], decreased spontaneous tumorigenesis, decreased cardiac and renal pathologies [78], and the overall improvement in lifespan of these mice. The MST1-mediated BECLIN1 phosphorylation that enhances the BECLIN1:BCL-2 interaction to reduce autophagy also consequently reduces the interaction between BCL-2 and BAX, causing BAX activation and stimulation of apoptosis during myocardial infarction [73]. Targeting the BECLIN1:BCL-2 interaction here might be advantageous in cases of cardiac dysfunction.

However, the benefits to increased autophagy are context-dependent and there are instances where blocking autophagy may be advantageous. One example is how upregulated autophagy in established tumours can assist their survival by providing molecular substrates in nutrient-depleted and hypoxic environments, promoting mitochondrial homeostasis, protecting against apoptosis, and enabling survival during the epithelial to mesenchymal transition and metastatic process [79,80,81]. Notably, the BH3-mimetic ABT-737, the predecessor to the clinically approved chemotherapeutic agent Venetoclax, can disrupt the BECLIN1:BCL-2/BCL-XL interaction to promote autophagy. Thus, this “off-target”—more accurately, “off-function”—effect during cancer treatment might act as an undesirable drug resistance mechanism [82].

### 4.2. Phosphorylation Events beyond the BECLIN1 BH3 Domain

Phosphorylation events that influence autophagic activity are not restricted to the BECLIN1 BH3 domain and regulation of BCL-2 binding affinity. The N-terminal, intrinsically disordered region of BECLIN1 is also a hotspot for kinases in nutrient sensing pathways, and activation of the associated kinases typically stimulates autophagy after their phosphorylation in this region. Notably, the same sites are often targeted by multiple kinases (Figure 3).

Perhaps the most important of all these kinases are the Unc-51 like kinase 1 (ULK1) and AMP-activated protein kinase (AMPK), known players in the classical molecular pathways of macroautophagy upstream of BECLIN1 that act to induce nucleation of the autophagosome. ULK1 phosphorylates BECLIN1 at S15 and S30 [83,84], whereas AMPK phosphorylates S93 and S96 in response to glucose starvation [85]. Similarly, several other kinases have been identified to phosphorylate this N-terminal region, leading to autophagy induction in response to various environmental stimuli. Hypoxia and glutamine deprivation can stimulate phosphoglycerate kinase 1 (PGK1) to phosphorylate BECLIN1 at S30 [86]. In particular, S90 appears to be the favoured target for most other kinases identified so far, including DAPK3 [87], mitogen activated protein kinase 2 and 3 (MAPK2/MAPK3, [88]), and calcium/calmodulin-dependent protein kinase II (CaMKII, [89]). It is also possible for the phosphorylation event at S90 to be reversed. Phospho-S90 induced in skeletal muscle after starvation can be dephosphorylated by the phosphatase PP2A, indicating that these phosphorylation modifications can be fine-tuned to carefully regulate autophagy levels [87]. Interestingly, in addition to a role in autophagy, AMPK-mediated phosphorylation of BECLIN1 at S90, S93, and S96 was demonstrated to enable BECLIN1 complex formation with Solute carrier family 7 member 11 (SLC7A11), driving lipid peroxidation and ferroptosis, highlighting another mechanism of crosstalk between autophagy and cell death regulated by BECLIN1 [90,91].

Notably, several studies have shown that phosphorylation of BECLIN1 might only occur within specific BECLIN1-containing complexes. Targeting of S90 and S93 appears to be dependent on ATG14, suggesting that only the BECLIN1 in C1 is phosphorylated at these sites [92]. ULK1-driven S30 phosphorylation is also only found in C1 [83]. To date, no C2-specific BECLIN1 phosphorylation events and their impact(s) on autophagy have yet been uncovered.

### 4.3. Phosphorylation Events Mediated by Oncogenic Kinases

Unlike the *N*-terminal intrinsically disordered region, the majority of PTMs in the ECD/CC domains appear to be associated with decreased autophagy, likely as a result of disrupting BECLIN1 interactions with the other C1 and C2 subunits. These are mostly mediated by oncogenic kinases. Phosphorylation at Y233 and Y352 by the chromosome fusion product BCR-ABL kinase reduced BECLIN1 interaction with UVRAG, VPS15, VPS34, and ATG14 whilst increasing RUBICON binding [93]. Oncogenic EGFR can similarly phosphorylate BECLIN1 at these same sites, plus Y229. This appears to promote BECLIN1 homodimerisation and BCL-2 association, as well as RUBICON binding, whilst inhibiting PI3K complex formation, all of which are autophagy-suppressive and associated with enhanced tumorigenesis in a non-small cell lung carcinoma xenograft model [94]. Another kinase, focal-adhesion kinase (FAK), though not strictly an oncogene but upregulated in many cancers, also targets the Y233 site, appearing to supress ATG14 interaction and decrease autophagy in cardiomyocytes to cause hypertrophy [95]. In addition, the receptor tyrosine kinase HER2 can engage BECLIN1 at the plasma membrane via the BECLIN1 ECD domain. Although it was not investigated whether this mediates phosphorylation on BECLIN1, the interaction was still shown to be autophagy-suppressive and was linked to increases in both phospho-HER2 and phospho-AKT levels (indicative of active cell signalling pathways, [96]). Consistent with these results, activated AKT was found to phosphorylate BECLIN1 at S234 and S295, resulting in supressed autophagy and tumour suppressor functions [97]. AKT-phosphorylated BECLIN1 has increased interaction with the adaptor protein 14-3-3 and the intermediate filament vimentin, supposedly sequestering it at the cytoskeleton to prevent autophagy [97]. Taken together, autophagy disruption mediated by oncogenic proteins inhibiting BECLIN1 function is likely one of the mechanisms by which these proteins can mediate their tumour maintenance and/or promoting effects. Currently, the only known exception to the inhibition of autophagy through BECLIN1 phosphorylation in the ECD/BARA domain is the phosphorylation of T388 by AMPK, which, instead, activates autophagy [98].

### 4.4. BECLIN1 Regulation via Ubiquitination Events

A universal way in which cells can regulate their homeostasis and general or tissue-specific functions is by the up- or downregulation of protein levels. BECLIN1 is no exception to this strategy and can be classically targeted for degradation via the ubiquitin–proteasomal system (Figure 4).

Ubiquitination events resulting in BECLIN1 protein degradation are typically K48-linked polyubiquitinations that are associated with decreased autophagy and linked to negative effects on immunity and cancer prognosis. Reduced BECLIN1 levels caused by K48-linked ubiquitination mediated by the E3 ligase ring finger protein 216 (RNF216) in toll-like receptor (TLR)-stimulated macrophages increase susceptibility to bacterial infection [99] and also promote colon tumour progression [100], effects credited to decreased autophagy. Similarly, the CULLIN 3 (CUL3) E3 ligase complex promotes K48-linked polyubiquitination of BECLIN1 and is associated with reduced autophagy, promoting the proliferation of breast and ovarian tumour cells [101]. S-phase kinase-associated protein 2 (SKP2)-mediated K48-ubiquitination of BECLIN1 at K402 also reduces autophagy and is linked to the virulence of MERS-CoV and SARS-CoV-2 infections [102,103]. This can be reversed with BECLIN1 stabilisation through pharmacological SKP2 inhibition via the drug niclosamide [103]. Unlike these E3 ligases, neural-precursor cell-expressed developmentally downregulated 4 (NEDD4) instead induces K11- and K63-linked polyubiquitination, though the consequence is the same as with K48 modification, delivering BECLIN1 for proteasomal degradation [104].

BECLIN1 is one of the few proteins targeted by multiple ubiquitination cascades [105] and can also be stabilised by ubiquitination events. The pattern emerging here is that K63-linked ubiquitination mediated by E3-ligases improves BECLIN1 scaffolding function for PI3K complex subunits. The E3 ligase TRAF6 induces K63-linked polyubiquitination at BECLIN1 K117, facilitating TLR4-stimulated autophagy in macrophages. This can be countered by the deubiquitinating enzyme (DUB) A20 [106]. Strategic placement of this ubiquitination event in the BH3 region suggests that it could displace BECLIN1 from BCL-2 to drive autophagy. Likewise, the AMBRA1-containing Ring box 1/Cullin 4 (RBX1/CUL4)-E3 ligase complex is responsible for the K63-linked polyubiquitination of BECLIN1 at K437 under starvation conditions. This ubiquitination event, which can be suppressed by the endosomal protein Wiscott-Aldrich syndrome protein (WASP) and SCAR homologue (WASH), putatively blocking ligase binding and enhancing VPS34 binding and hence autophagy [107]. TRIM50 also mediates K63-linked ubiquitination at an unknown site with pro-autophagic outcomes. This is proposed to be due to the promotion of the BECLIN1:ULK1 interaction and possibly increased autophagy activating phosphorylation of BECLIN1 as a result, although this is not experimentally confirmed [108]. Opposing its BECLIN1 degradative function mentioned above, NEDD4 was also found to elevate BECLIN1 stability through K6- and-K27 linked ubiquitination, an event that enhanced intracellular bacterial pathogen clearance via autophagy, although the mechanism for this is also unclear [109].

In contrast to the E3 ligase-mediated regulation of BECLIN1 protein levels, DUBs can remove ubiquitination events, reversing the degradation or stabilisation of BECLIN1. Other than the aforementioned DUB A20-mediated removal of TRAF6 ubiquitination, DUBs to date have been found to remove degradative BECLIN1 ubiquitination events to stabilise BECLIN1 levels, leading to increased autophagic flux. These include the DUBs USP10, USP13, and UPS19, where the latter two were reported to specifically target K11-linked chains (at K437 for USP19) and increase autophagy [110,111]. Similarly, USP14 removes K63-linked ubiquitinations from BECLIN1, with a loss of this protein suppressing autophagy [112]. In addition to the immune- and cancer-related functions of BECLIN1 ubiquitination events, the stabilisation of BECLIN1 by K48-linked deubiquitination at K402 by the DUB Ataxin-3 is important for maintaining autophagy in brain models of Huntington’s disease [113].

### 4.5. BECLIN1 Regulation via Caspases and Other Proteases

Although not as heavily studied as phosphorylation and ubiquitination, another manner in which BECLIN1 activity is regulated is through proteolytic cleavage events. A mechanism of crosstalk between autophagy and apoptosis, in addition to the BECLIN1:BCL-2 interaction, is the cleavage of BECLIN1 by apoptotic caspases, leading to its functional inactivation and reduced autophagy. In response to various stimuli, including tumour necrosis factor-related apoptosis-inducing ligand (TRAIL), caspases-3 and -8 can cleave BECLIN1 at D124 and D149 or D133 and D146, respectively [114,115,116,117,118]. BECLIN1 fragments resulting from D149-targeted caspase-3 cleavage mislocalise to the nucleus, leading to reduced interaction with VPS34 [116]. It was also found that the C-terminal BECLIN1 fragment (residues 150–450) localises to the mitochondria and is capable of releasing the pro-apoptotic proteins cytochrome *c* and HtrA2/Omi, possibly providing a positive feedback loop for apoptosis [117]. Together, these data suggest mechanisms by which autophagy can be switched off during apoptosis and BECLIN1 instead deployed to amplify cell death. Physiologically, these mechanisms are linked to a neuroregulatory role, as increased BECLIN1 cleavage products were found in the brains of Alzheimer’s disease patients [119]. The enforced expression of the BECLIN1 C-terminal cleavage product (residues 148–450) sensitised neuronal cells to death following receipt of excitotoxic insults and could be prevented in vivo using caspase inhibitors [119]. Outside of the family of apoptosis effector caspases, calpain, a calcium-dependent caspase, was also shown to cleave BECLIN1 and give rise to a 40 kDa C-terminal product, sensitising cells to death by oxidative stress [120].

## 5. Concluding Remarks

As discussed, BECLIN1 is a protein known for its key role in regulating autophagy and other membrane trafficking processes as an essential member of the Class III PI3K complexes. Without BECLIN1, the stability of other members of these complexes (ATG14 and UVRAG) is compromised [12,42]. Protein–protein interactions, phosphorylation, ubiquitination, cleavages, and other events instigated on BECLIN1 by a diverse array of cell pathways all affect its ability to function as the signal integrating scaffold in these complexes, making BECLIN1 a significant site of crosstalk between autophagy, apoptosis, and cell proliferation.

Knowledge of these regulatory processes is useful as they could potentially be exploited for therapeutic benefit in diseases where BECLIN1 has been implicated, such as cancers where decreased BECLIN1 expression is associated with poorer prognosis and survival [121]. Similarly, decreased BECLIN1 expression is seen in the brains of Alzheimer’s disease patients [122]. One point to note here is that the role of BECLIN1 as a tumour suppressor is somewhat contentious due to its low somatic and germline mutation status across multiple cancer types [123,124]. Rather, it has been suggested that, due to its close chromosomal proximity to the breast cancer tumour suppressor BRCA1, it could be a passenger mutation rather than the driver mutation and simply co-deleted with BRCA1 [123]. Regardless, activation of BECLIN1-associated pathways is of interest as a therapeutic modality; however, whether manipulation of BECLIN1 and its interactome could be advantageous in this regard has not been widely explored. Drugs that have been developed to target the BCL-2 pro-survival protein BH3 binding site (the aforementioned “BH3-mimteics”) have been shown to enhance autophagy by disrupting the BECLIN1:BCL-2/BCL-XL interaction, though the exact mechanism(s) involved here has also been disputed [91,125]. More recently, specific putative “BECLIN1-mimetics” that function in the same way have been reported [126]. Attempts to develop agents against BECLIN1 or its other binding partners have led to the generation of the Tat-Beclin1 peptide. This modified BECLIN1-derived peptide, linked to the cell penetrating TAT transduction domain of HIV-1, is capable of enhancing autophagy through its interaction with Golgi-associated plant pathogenesis-related protein-1 (GAPR1), a negative regulator of autophagy, and was shown to be beneficial in clearing polyglutamine expansion aggregates and some viral infections [127]. Hence, this apparently promising approach to targeting BECLIN1-regulated pathways may have some future therapeutic utility.

Finally, many of the cellular and physiological consequences of regulatory modifications on BECLIN1 have been examined from the standpoint of its role in autophagy, often overlooking the role it plays in other trafficking pathways, which now have confirmed importance in neurological health and are essential to proper organismal development. Future studies on BECLIN1, looking not only at its role in autophagy but also its role in other membrane trafficking pathways and whether there are C1- or C2-specific modifications, might unveil important mechanisms of cellular function that explain its already established physiological roles, and the others yet to be discovered.

## Figures and Tables

**Figure 1 cells-10-01522-f001:**
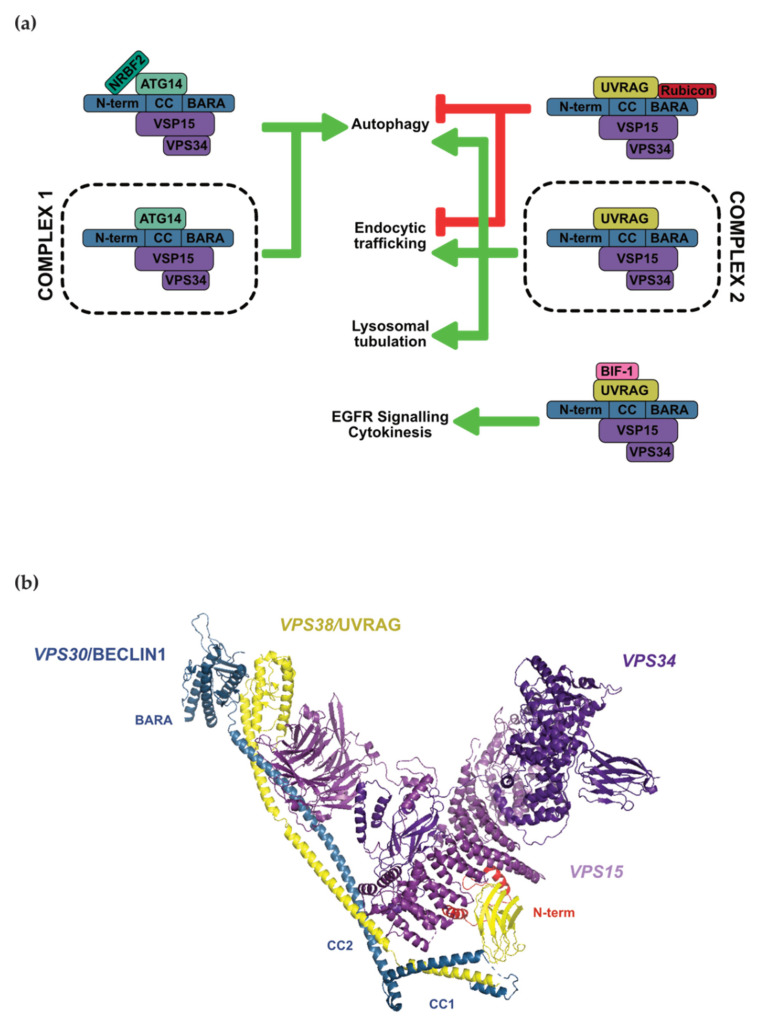
(**a**) PI3K Class III complexes, subcomplexes, and their associated cellular functions. BECLIN1 is indicated in blue with its protein domains listed. Green arrows indicate functions promoted by complexes, whereas red indicates pathways inhibited by complexes. (**b**) Yeast C2 crystal structure resolved to 4.40 Å by X-ray diffraction: *VPS30* (BECLIN1 homolog, blue), *VPS38* (UVRAG homolog, yellow), *VPS15* (purple), *VPS34 (*deep purple). Red helices have been tentatively assigned to the Beclin1 N-terminal region that contains the BH3 domain (PDB ID: 5DFZ).

**Figure 2 cells-10-01522-f002:**
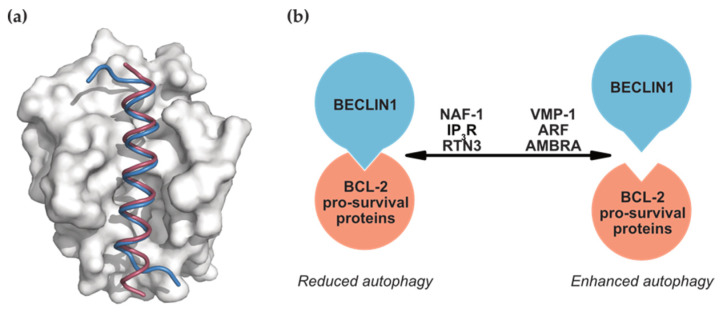
(**a**) Structure of BCL-XL in complex with the BH3 domains of the pro-apoptotic BH3-only protein BIM (blue, PDB: 3FDL) [61] and BECLIN1 (pink, 2P1L) [57]. Both BH3 domains occupy the same binding site. (**b**) Schematic of proteins that competitively bind BECLIN1 and/or BCL-2 to promote or inhibit this interaction, thereby reducing or enhancing autophagy, respectively.

**Figure 3 cells-10-01522-f003:**
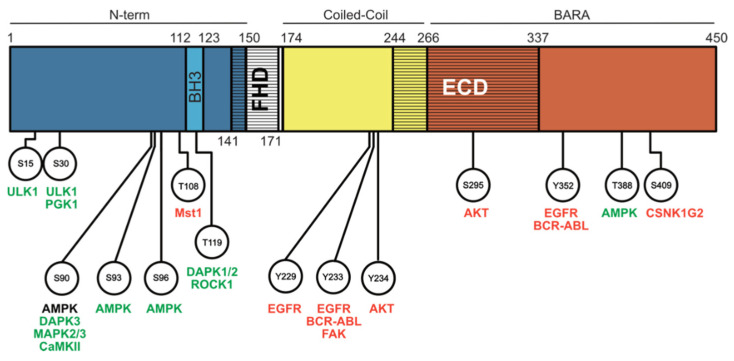
Phosphorylation events that regulate BECLIN1. Kinases that induce autophagy are indicated in green, and those with inhibitory effects are indicated in red. Black indicates a kinase/phosphorylation site of unclear effect on autophagy.

**Figure 4 cells-10-01522-f004:**
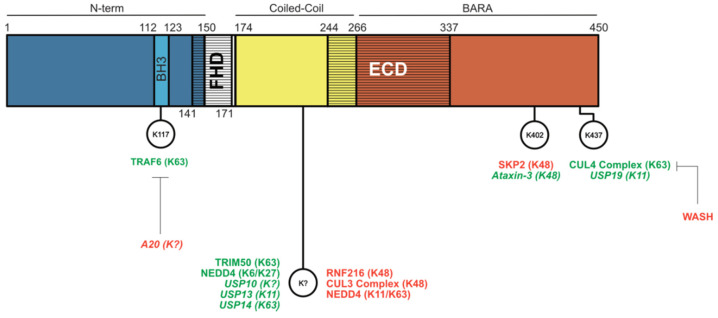
Ubiquitination events that regulate BECLIN1. E3 ligases that stabilise BECLIN1 protein levels and/or promote autophagy are indicated in green, whereas those with degradative effects on BECLIN1 that downregulate autophagy are indicated in red (note: WASH is the only non-E3 ligase indicated here). DUBs are italicised and also follow the green and red colour code to indicate their pro- or anti-autophagic effects.

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
