# Peer review of "BECLIN1: Protein Structure, Function and Regulation"

_cells, 2021, doi:10.3390/cells10061522_

Round 1
Reviewer 1 Report
The review by Tran et al “BECLIN1: PROTEIN STRUCTURE, FUNCTION and REGULATION.” comprehensively described the molecular mechanisms regulating BECLIN1 function and the physiological consequences of these interactions. This review is well written and very informative. I have only minor suggestions in regard to the figures. I would suggest increasing the font size of the text in Figure 1a, Figure 2 and Figure 3. Also, it would be good to include an additional schematic figure corresponding to the section 4.1 BECLIN1:BCL-2 (similar to that in Figure 1a).
Author Response
We thank the reviewer for their positive comments on our manuscript and useful suggestions as to how to improve it. Accordingly, we have made the following changes:
1) Font sizes have been increased in Figures 1a, 2 (now Figure 3) and 3 (now Figure 4).
2) We have now included an additional figure, as suggested (Figure 2a,b). This shows (a) the structure of BCL-2 protein complexes with BECLIN1 BH3 and BIM BH3 showing the conservation of this mode of binding between apoptosis and autophagy pathways; and (b) a schematic of the various other binding proteins that either positively or negatively regulate this interaction.
Reviewer 2 Report
This review was a delight to read and put together by experts in the autophagy field.
This was a concise summary of the complex literature on the enigmatic functions of Beclin in a multitude of signalling pathways and its regulation to finely balance autophagy induction. A great review for those who need a broad and succinct overview of Beclin signalling and function, with nice illustrations of Beclin complexes to summarise its many roles.
The authors have also included some perspective on the signalling and some physiological impacts of fine-tuning Beclin regulation. The only thing I would perhaps suggest (although not highly essential for its publication), is to have a final paragraph describing what is known about Beclin in human disease and cancer, the known somatic mutations and altered expression levels that may impact cancer progression. This will provide some perspective on how targetting Beclin in cancer may be beneficial.
Author Response
We thank the reviewer for their positive comments on our manuscript and suggestions on how we could improve it. Accordingly, we have now added the following paragraph to the "Concluding Remarks" section discussing role of BECLIN1 in disease and the potential for targeting it (or its binding partners) for therapeutic benefit.
"Knowledge of these regulatory processes is useful as they could potentially be exploited for therapeutic benefit in diseases where BECLIN1 has been implicated, such as cancers where decreased BECLIN1 expression is associated with poorer prognosis and survival [121]. Similarly, decreased BECLIN1 expression is seen in the brains of Alzheimer’s disease patients [122]. One point to note here is that the role of BECLIN1 as a tumour suppressor is somewhat contentious due to its low somatic and germline mutation status across multiple cancer types [123, 124]. Rather it has been suggested that due to its close chromosomal proximity to the breast cancer tumour suppressor BRCA1, it could be a passenger mutation rather than the driver mutation and simply co-deleted with BRCA1 [123]. Regardless, activation of BECLIN1-associated pathways is of interest as a therapeutic modality, however, whether manipulation of BECLIN1 and its interactome could be advantageous in this regard has not been widely explored. Drugs that have been developed to target the BCL-2 pro-survival protein BH3-binding site (“BH3-mimteics”) have been shown to enhance autophagy by disrupting the BECLIN1:BCL-2/BCL-XL interaction, though the exact mechanism(s) involved here has also been disputed [91, 125]. More recently, specific putative “BECLIN1-mimetics” that function in the same way have been reported [126]. Attempts to develop agents against BECLIN1 or its other binding partners has led to the generation of the Tat-Beclin1 peptide. This modified BECLIN1-derived peptide linked to the cell penetrating TAT transduction domain of HIV-1, is capable of enhancing autophagy through its interaction with Golgi-associated plant pathogenesis-related protein-1 (GAPR1), a negative regulator of autophagy), and was shown to be beneficial in clearing polyglutamine expansion aggregates and some viral infections [127]. Hence, this apparently promising approach to targeting BECLIN1-regulated pathways may have some future therapeutic utility."
Reviewer 3 Report
Tran et al., submitted the review entitled “BECLIN1: PROTEIN STRUCTURE, FUNCTION AND REGULATION”. This an excellent review, which covers most of the aspects of Beclin1. The authors have clearly described Beclin1 interactions either in the presence or absence of PTMs. I think the manuscript does not need any additional inputs. As such, it acceptable for publication.
Hereby I endorse the manuscript for publication in its present form.
Author Response
We thank the reviewer for their positive comments on our manuscript and accordingly have not made any further changes except for those suggested by the other reviewers.